# Cholesterol and Sphingomyelin Polarize at the Leading Edge of Migrating Myoblasts and Involve Their Clustering in Submicrometric Domains

**DOI:** 10.3390/biom13020319

**Published:** 2023-02-07

**Authors:** Juliette Vanderroost, Noémie Avalosse, Danahe Mohammed, Delia Hoffmann, Patrick Henriet, Christophe E. Pierreux, David Alsteens, Donatienne Tyteca

**Affiliations:** 1de Duve Institute, UCLouvain, 1200 Brussels, Belgium; 2Louvain Institute of Biomolecular Science and Technology (LIBST), UCLouvain, 1340 Ottignies-Louvain-la-Neuve, Belgium

**Keywords:** membrane lateral heterogeneity, Airyscan microscopy, atomic force microscopy, GM1 ganglioside, focal adhesion, actin cytoskeleton, cell polarization, FRAP, membrane fluidity

## Abstract

Myoblast migration is crucial for myogenesis and muscular tissue homeostasis. However, its spatiotemporal control remains elusive. Here, we explored the involvement of plasma membrane cholesterol and sphingolipids in this process. In resting C2C12 mouse myoblasts, those lipids clustered in sphingomyelin/cholesterol/GM1 ganglioside (SM/chol/GM1)- and cholesterol (chol)-enriched domains, which presented a lower stiffness than the bulk membrane. Upon migration, cholesterol and sphingomyelin polarized at the front, forming cholesterol (chol)- and sphingomyelin/cholesterol (SM/chol)-enriched domains, while GM1-enriched domains polarized at the rear. A comparison of domain proportion suggested that SM/chol- and GM1-enriched domains originated from the SM/chol/GM1-coenriched domains found at resting state. Modulation of domain proportion (through cholesterol depletion, combined or not with actin polymerization inhibition, or sphingolipid synthesis inhibition) revealed that the higher the chol- and SM/chol-enriched domains, the higher the myoblast migration. At the front, chol- and SM/chol-enriched domains were found in proximity with F-actin fibers and the lateral mobility of sphingomyelin in domains was specifically restricted in a cholesterol- and cytoskeleton-dependent manner while domain abrogation impaired F-actin and focal adhesion polarization. Altogether, we showed the polarization of cholesterol and sphingomyelin and their clustering in chol- and SM/chol-enriched domains with differential properties and roles, providing a mechanism for the spatial and functional control of myoblast migration.

## 1. Introduction

Skeletal myogenesis is an important process of embryonic development that allows the formation of skeletal muscle tissue [1]. It also occurs in adulthood during tissue homeostasis, muscle regeneration after injury, and muscle hypertrophy after exercise [2,3]. This process can be divided into distinct steps. Briefly, a wide variety of signaling molecules will induce the specification of mesodermic-derived structures into myogenic progenitors, which start to express specific myogenic transcription factors. These progenitors will then commit to the myogenic program by differentiating into myoblasts and proliferating. Myoblasts will migrate towards each other, align, and fuse first to form nascent binucleated myotubes and then elongated and multinucleated myotubes. These latter will finally grow into mature myofibers. A small number of committed progenitors, called satellite cells, will remain closely apposed to the myofiber in a quiescent state. They will be able to differentiate into myoblasts and migrate and fuse with the preexistent muscle when stimulated [4,5].

Myoblasts are generally described to preferentially migrate in a single-cell mesenchymal way in vitro [6,7,8], even though modulation of migration modes and actin organization can occur, presumably in response to changes in cell confinement and adhesion [9,10,11]. This mode of migration is a well-organized and polarized process usually described in four steps [12]: (i) formation of lamellipodia through branched actin filament polymerization at the leading edge; (ii) adhesion of the leading edge through the formation of new focal adhesions (FAs), which connect the cell cytoskeleton to the extracellular matrix and initiate various intracellular signaling cascades primarily involved in the reorganization of the actin cytoskeleton [13]; (iii) translocation of the cell body induced by myosin-dependent contractile forces on thick and stable actin cables linked to FAs; and (iv) cell retraction of the trailing edge through actin depolymerization and FAs disassembly. From a molecular point of view, the cytoskeleton dynamics upon migration are mainly controlled by small GTPases from the Rho family, such as Rac, Cdc42, and RhoA [14]. The control of actin dynamics by GTPases is similarly important for later myoblast fusion through the formation of actin-propelled invasive protrusions at the site of fusion [15,16,17].

To face the mechanical forces driven by the constant remodeling of the actin cytoskeleton, the plasma membrane (PM) of migrating and fusing myoblasts must display sufficient deformability [18,19]. Such deformability is controlled by the cholesterol (chol) content and biophysical properties, i.e., fluidity, curvature, and transversal asymmetry [20,21]. Although data supporting the role of PM biophysical properties in myoblast migration are lacking, the contribution of membrane fluidity and transversal asymmetry to myoblast fusion is supported by little evidence. In L6 skeletal muscle cells, the ratios of chol to phospholipid and saturated to unsaturated fatty acids decrease upon myoblast differentiation [22]. Moreover, chol depletion enhances embryonic chick skeletal muscle cell fusion [23], and sphingomyelin (SM) levels in the PM decrease upon differentiation of C2C12 myoblasts [24]. Finally, phosphatidylserine flip-flop from the internal to the external PM leaflet of C2C12 myoblasts contributes to myotube formation [25].

As an additional layer of complexity, PM biophysical properties can vary locally as a consequence of specific lipid clustering in domains [26]. In different cellular systems, lipid domains at the outer PM leaflet have been shown to contribute to cell migration: (i) chol-enriched submicrometric domains at the surface of breast cancer cell lines participate in invadopodia formation and cancer cell invasion [27]; (ii) SM- and chol-enriched submicrometric domains at the basal side of keratinocytes are involved in their movement [28]; and (iii) GM1 ganglioside-enriched domains polarize at the leading edge of migrating ECV304 human epithelial cells [29] or at the rear of T lymphocytes [30]. The polarization at the outer PM leaflet echoes the preferential distribution of phosphatidylinositol-4,5-biphosphate (PI4,5P_2_) and phosphatidylinositol-3,4,5-trisphosphate (PI3,4,5P_3_) at the inner PM leaflet of migrating fibroblasts. PI3,4,5P_3_ accumulation at the leading edge favors the formation and activation of Rac1 nanoclusters, leading to actin polymerization and fibroblast migration [31,32]. However, the relevance of lipid domains at the myoblast PM and their role in myoblast migration are poorly understood.

In the present study, we build on our evidence of lipid domains at the surface of red blood cells (RBCs) and breast cancer cell lines and their roles in RBC deformation and breast cancer cell invasion [27,33,34,35,36,37] to address those issues. Using the well-characterized murine C2C12 myoblast cell line [38], we specifically investigated whether SM, chol, and GM1 can cluster into lipid domains at the myoblast cell surface. These three lipids were selected based on their enrichment in the outer PM leaflet [39] and because they play key pathophysiological roles. We then explored lipid domain diversity and biophysical properties (e.g., fluidity), their implication in myoblast spontaneous and oriented migration, and the underlying mechanism.

## 2. Materials and Methods

### 2.1. Cell Culture and Chemical Treatments

C2C12 myoblasts were grown at 37 °C with 5% CO_2_ in Dulbecco’s Modified Eagle Medium (DMEM, Gibco) containing 4.5 g/L D-glucose and 25 mM HEPES and supplemented with 10% Fetal Bovine Serum (FBS), penicillin (100 U/mL) and streptomycin (100 µg/mL). For experiments in resting conditions, cells were grown overnight on 2 cm² glass coverslips at 10,000 cells/cm^2^, reaching ~50% confluency within 24 h. For experiments on migrating cells, cells were grown on 0.44 cm^2^ Ibidi Culture-Insert 2 Well dishes at 20,000 cells/cm^2^, which allowed them to reach ~80% confluency at the migration initiation time. To deplete chol, cells were pre-incubated in a serum-free medium containing 2.5–10 mM methyl-β-cyclodextrin (mβCD, Sigma-Aldrich, St. Louis, MO, USA) for 30 min at 37 °C. To subsequently replete chol, depleted cells were incubated in a serum-free medium containing 2.5 mM mβCD charged with cholesterol-water soluble (mβCD:chol, Sigma-Aldrich) in a 6:1 molar ratio for 1 h at 37 °C. To inhibit sphingolipid synthesis, cells were pre-incubated for 48 h at 37 °C with 7.5–30 µM fumonisin B1 in a serum-containing medium (FB1, Sigma-Aldrich). To inhibit actin polymerization, cells were pre-incubated in a serum-free medium containing 0.5–1 µM cytochalasin D (cytoD, Sigma-Aldrich) for 30 min at 37 °C. To simultaneously deplete chol and inhibit actin polymerization, cells were pre-incubated in a serum-free medium containing both 5 mM mβCD and 0.5 µM cytoD for 30 min at 37 °C. FB1 and cytoD were maintained during migration experiments.

### 2.2. Cholesterol Content Determination

Cells were grown overnight in T75 flasks, then pre-treated with mβCD, cytoD, a combination of both drugs, or with FB1, as explained above. If necessary, treatment was followed by chol repletion with mβCD:chol complexes. Cells were lysed, and total cell chol content was extracted as described in [33]. Total and membrane chol contents were assessed using the Amplex™ Red Cholesterol Assay Kit (Invitrogen, Waltham, MA, USA) upon the addition of chol esterase or not. 

### 2.3. Sphingomyelin Content Determination

Cells were grown overnight in T75 flasks, then pre-treated with mβCD or FB1. Cells were lysed, total lipid content was extracted, then separated by thin layer chromatography (TLC) and revealed as previously described [27].

### 2.4. Confocal Live Cell Imaging of Plasma Membrane Sphingomyelin, Cholesterol, GM1 Ganglioside, Non-Specific Membrane Staining, and Actin on Resting and Migrating Myoblasts

Cells seeded on glass coverslips or Ibidi chambers were washed twice with ice-cold serum-free DMEM and then labeled at 4 °C (except otherwise stated) with commercially available lipid analogs or fluorescent toxin fragments produced as previously described [34,37]. For migration experiments, cells were first allowed to migrate for 5 h at 37 °C, then labeled at 4 °C with a 2-fold lower probe concentration than for experiments at the resting state to adjust for the 2-fold lower total number of seeded cells. Thus, for SM labeling, cells were incubated with 0.3–0.6 µM BODIPY-SM (Invitrogen) or with 0.75–1.5 µM mCherry-lysenin toxin fragment (hereafter named lysenin) for 15 min in medium containing 1 mg/mL fatty acid-free bovine serum albumin (FFA BSA, Sigma-Aldrich) (DMEM/FFA BSA). For chol labeling, cells were incubated with 1.25–2.5 µM TopFluor-Chol for 20 min in 3.75–7.5 µM mβCD or with 1–2 µM mCherry-theta toxin fragment (hereafter named theta) for 20 min in DMEM/FFA BSA. For GM1 labeling, cells were incubated with 0.5–1 µM BODIPY-GM1 (Invitrogen) for 15 min or with 5–10 µg/mL Alexa Fluor 647-conjugated Cholera toxin B subunit (CTxB, Invitrogen) for 30 min in DMEM/FFA BSA. To remove BODIPY-lipids from the PM, cells were incubated twice with 5% BSA (Biowest, Riverside, MO, USA) for 15 min at 4 °C. For the non-specific PM staining, cells were incubated with 10 µg/mL FM4-64X (Invitrogen) for 10 min in a serum-free medium. For actin labeling, cells were incubated with 1 µM SiR-actin probe (Spirochrome, Stein am Rhein, Switzerland) for 30 min at 37 °C. In cases of double- or triple-labeling, probes were simultaneously incubated with the cells except for (i) SiR-actin, which was first incubated at 37 °C and then with lipid probes at 4 °C, and (ii) double-labeling of the same lipid using two different probes, which was carried out sequentially with a toxin fragment followed by a lipid analog. After labeling, cells were washed twice with ice-cold serum-free medium, except for those labeled with FM4-64X, which were immediately visualized. For resting experiments, coverslips with labeled living cells were transferred upside down into medium-filled LabTek chambers and visualized with a Zeiss Cell Observer Spinning Disk (COSD) confocal microscope using a plan-Apochromat 63x/1.4 water-immersion objective. For migration experiments, labeled living cells were directly observed in Ibidi Culture-Insert 2 Well dishes with a Zeiss Laser Scanning Microscope LSM980 Airyscan 2 confocal microscope using a plan-Apochromat 63x/1.4 oil-immersion objective. Images were taken in X-Y basal plans and/or X-Z orthogonal sections.

### 2.5. Quantifications of Surface Lipid Labeling, Lipid Colocalization, and Domain Proportion and Abundance

The intensity of surface lipid labeling was analyzed on X-Y basal images using ImageJ software. Images were converted to an 8-bit grayscale, and the mean gray value (MGV) of the labeling was extracted from the entire surface for resting cells or from the front and the posterior end for migrating cells. Background MGV was then subtracted. Data are presented as a percentage of control for fluorescence intensity on resting cells and as an MGV ratio (front vs. rear) for lipid polarity analysis on migrating cells. 

Lipid colocalization was also analyzed on basal X-Y images from double-labeled cells using green or red colocalization coefficients (CC) generated by the colocalization tool of the ZEN software (Zeiss, Jena, Germany). The green or red CC is the colocalized pixel count divided by the sum of the colocalized and the green or red channel pixel counts. 

Lipid domain proportion was determined based on their enrichments in SM, chol, or GM1 using the profile tool of the ZEN software (Zeiss). Virtual lines were manually drawn across as many lipid domains as possible on the entire cell periphery of basal sections of triple-labeled cells, generating graphs of the fluorescence intensity along the profiles. Thresholds were set to differentiate peaks from background signals. These thresholds were different for the green, red, and far-red channels but remained identical between different conditions of a single experiment. Peaks in fluorescence intensity were then manually quantified to determine the proportion of the seven differentially enriched lipid domains identified based on the enrichment of one, two, or three lipids. Data were expressed in % of the total lipid domains number. 

To determine the total lipid domain abundance relative to the cell surface, the total number of domains quantified in the “proportion analysis” was added, then divided by the cell surface and expressed as the number of lipid domains/2000 µm², which represents the average myoblast surface.

### 2.6. Atomic Force Microscopy and Quantification

As previously described, atomic force microscopy (AFM) tips derivatized with lysenin- and theta-toxin fragments were obtained [36,40]. AFM images of confluent layers of C2C12 cells were acquired using an AFM (Bioscope Resolve, Bruker, Billerica, MA, USA) operated in the PeakForce QNM mode (Nanoscope software v9.2) and coupled to an inverted epifluorescence microscope (Zeiss Observer Z.1) or a confocal laser scanning microscope (Zeiss LSM980). A 40x oil objective (NA = 0.95) was used. The AFM was equipped with an X-Y 150 μm piezoelectric scanner and a cell-culture chamber, allowing it to control the temperature, humidity, and CO_2_ concentration [41]. Overview images of cell surfaces (20–80 μm^2^) were recorded at imaging forces of 500–750 pN using PFQNM-LC probes (Bruker) having tip lengths of 17 μm, tip radii of 65 nm, and opening angles of 15°. All fluorescence microscopy and AFM imaging experiments were conducted under cell-culture conditions using the combined AFM and fluorescence microscopy chamber at 37 °C in the medium. A gas mixture of synthetic air with CO_2_ (5%) at 95% relative humidity using a gas humidifier membrane (PermSelect silicone) was infused at 0.1 L/min into the microscopy chamber. The humidity was controlled using a humidity sensor (Sensirion, Stäfa, Switzerland). Cantilevers were calibrated using the thermal noise method [42], yielding values ranging from 0.08 to 0.14 N/m for PFQNM-LC probes. The AFM tip was oscillated in a sinusoidal fashion at 0.25 kHz with a 750 nm amplitude in the PeakForce Tapping mode. The sample was scanned using a frequency of 0.125 Hz and 128 or 256 pixels per line (256 lines). AFM images were analyzed using the Nanoscope Analysis software (v1.9, Bruker) and ImageJ (v1.52e). Optical images were analyzed using Zen Blue software (Zeiss). Raw FD curves extracted from multiparametric FD-AFM maps were processed offline using the AtomicJ open-source software [43].

To correct for the tilt sometimes present in raw FD-curves, the baseline of the retraction curve was corrected using a 2nd-degree polynomial fit on the off-contact area. To extract Young’s modulus values, Hertz’s model for a sphere was fitted to the contact region of the retraction part of FD-curves [44]:(1)$$F^{2/3}={\left(\frac{4}{3}\;\frac{E}{\left(1-\nu^{2}\right)}\;\sqrt{R}\right)}^{2/3}\delta \;$$
where E is Young’s modulus, δ is the indentation depth, ν is the Poisson ratio, and R is the contact radius. We used a Poisson’s ratio value of 0.5. Young’s modulus was computed from the slope of Equation (1). Two indentation depth ranges were defined: ∆δ < 50 nm, corresponding to the PM contribution, and ∆δ > 50 nm, corresponding to the cell cortex. A correction for the effect of the substrate was implemented for the estimation of the PM Young’s modulus using a spherical indenter [45]:(2)$$F=\frac{16E}{9}\sqrt{R}\delta^{3/2}\left[1+0.884\alpha +0.781\alpha^{2}+0.386\alpha^{3}+0.0048\alpha^{4}\right]$$
where $\alpha =\frac{\sqrt{R\delta }}{h}$ and h is the thickness of the layer

The retraction part of FD-curves was also analyzed to measure specific unbinding events between chol-enriched assemblies and the theta-toxin derivatized AFM tip. An event was counted as specific when the minimum adhesion force was higher than 100 pN and the unbinding distance was more than 5 nm from the contact point. The noise level in raw FD curves was calculated by calculating the standard deviation from a linear fit of the off-contact part of the retraction curve. Typical noise level values were below 20 pN.

### 2.7. Spontaneous Migration Assay

Cells were grown overnight in Ibidi Culture-Insert 2 Well dishes and then pre-treated with mβCD, followed or not by chol repletion with mβCD:chol complexes, with cytoD, a combination of both treatments, or with FB1, as explained under Section 2.1. After treatment, the 2-well silicone insert was removed, thereby defining a cell-free gap suitable for spontaneous migration. Cells were left to migrate for 5 h in the serum-containing medium at 37 °C. Images were taken after 0 h and 5 h of migration with a wide-field fluorescence microscope, Observer Z.1 (10x objective). To calculate the migration distance, the gap width after 5 h, measured with ImageJ software, was subtracted from the width at time 0.

### 2.8. Oriented Migration Assay

Cells were grown in ThinCert™ Tissue Culture Inserts (8 µm pore size, Greiner, Mount Joy, PA, USA) with a serum-free medium in the bottom plate. Treatments were similar to spontaneous migration except for FB1: the first 24 h of treatment were conducted on cells seeded in 12-well plates; cells were then trypsinized and re-seeded in the inserts for the next 24 h of treatment. To induce oriented migration after treatment, inserts were filled with a serum-free medium, and bottom chambers contained either a serum-free medium (spontaneous migration) or 250 ng/mL insulin-like growth factor-1 (IGF-1, Sigma-Aldrich) diluted in DMEM/FFA BSA (stimulated migration). After 5 h of migration at 37 °C, migrating cells were fixed for 20 min with 4% paraformaldehyde (PFA), stained for 10 min with 0.5% crystal violet diluted in 4% PFA, and washed with phosphate-buffered saline (PBS). Cells that did not migrate through the insert were wiped out with cotton swabs. Ten images per insert were taken with a wide-field fluorescence microscope, Observer Z.1 (20x objective). Using the ZEN software, the total area of migrating cells was determined and divided by the area of a single cell (mean of 10 cells for each condition) to evaluate the number of cells per area. The oriented migration was calculated by subtracting the number of spontaneous migration from the number of stimulated migrating cells.

### 2.9. Focal Adhesion and Actin Cytoskeleton (Immuno)Fluorescence and Quantification

Seeded cells were washed with PBS, fixed for 30 min with 4% PFA, washed with PBS 3 times for 5 min, and permeabilized for 5 min with 0.3% Triton. Cells were then blocked for 1 h with 10% BSA and 3% milk diluted in PBS at room temperature (RT). Cells were next immunolabeled for 2 h with a primary anti-paxillin antibody (Merck Millipore, Burlington, MA, USA, dilution of 1:200), then incubated for 1 h at RT with an Alexa Fluor secondary antibody (ThermoFisher, Waltham, MA, USA, 1:500), Alexa Fluor Phalloidin (ThermoFisher, 1 U/mL), and Hoechst (ThermoFisher, 1:1000), all diluted in 1% PBS/BSA solution. Cells were then washed with PBS 3 times for 5 min. Coverslips containing resting cells were mounted with Dako Faramount Aqueous Mounting Medium on SuperFrost Plus microscope slides, and migrating cells were directly visualized in Ibidi chambers using the COSD confocal microscope with a plan-Apochromat 63x/1.4 water-immersion objective. The quantification of FA number and cell surface occupancy was performed as previously described [27,46]. The number of FAs was then normalized to the total cell surface, while surface occupancy distinguished the cell front, center, and rear. The quantification of actin cytoskeleton cell surface occupancy was performed using the same method. The quantification of FA and F-actin polarization was determined by the proportion of FA or F-actin area at the cell front, center, or rear compared to the total area of FA or F-actin.

### 2.10. Fluorescence Recovery after Photobleaching

Cells were grown in Ibidi Culture-Insert 2 Well dishes, then immediately labeled for SM (BODIPY-SM), chol (theta), or GM1 (CTxB) at RT for experiments on resting cells or were first left to migrate for 5 h at 37 °C for experiments on migrating cells. Fluorescence recovery after photobleaching (FRAP) was performed with a Zeiss Laser Scanning Microscope LSM980 Airyscan 2 confocal microscope using a plan-Apochromat 63x/1.4 oil-immersion objective. Several zones of 5 µm² were photobleached per myoblast: (i) the lipid domain and bulk membrane for resting cells; (ii) the lipid domain at the front (SM, chol) or at the rear (GM1), the bulk membrane at the front, and the bulk membrane at the rear for migrating cells. A control non-photobleached zone was also defined to check for fluorescence stability. Photobleaching parameters were set as follows: 5/8/10 iterations at 80/100/100% laser intensity for SM, GM1, and chol labeling, respectively. Images after photobleaching were taken every 8 s for 10 cycles for SM and every 10 s for 13 cycles for chol and GM1. Fluorescence recovery at time x was then defined as (fluorescence _time X_ − fluorescence _photobleaching_)/(fluorescence _time 0_ − fluorescence _photobleaching_). Data were then expressed as the mobile fraction determined by fitting the data with a one-phase association fitting (GraphPad Prism 8.0.2., GraphPad Software, San Diego, CA, USA).

### 2.11. Statistical Analyses

All statistical analyses were performed with GraphPad Prism 8.0.2. Data are represented as means of *n* independent experiments or means of x cells from *n* independent experiments ± SD. Statistical tests were performed when *n* ≥ 3, and tests were non-parametrical when *n* ≤ 10. To compare the effect of one treatment with a hypothetical mean of 100% representing the control, non-parametrical Kruskal-Wallis (unpaired data) or Friedman (paired data) followed by Dunn’s multiple comparisons test were used. To compare the mean of two different samples, a non-parametrical Wilcoxon signed-rank test or parametrical unpaired t-test was conducted. To compare more than two samples, a parametrical one-way ANOVA with Geisser–Greenhouse correction followed by Tukey’s multiple comparisons test was performed. Finally, to compare grouped samples, a parametrical two-way ANOVA with Geisser–Greenhouse correction followed by Sidak’s multiple comparisons test was conducted. Comparisons with the control value are indicated above the columns, while comparisons between two or more groups are indicated with bars on top of graphs. ns, non-significant; *, *p*-value < 0.05; **, *p*-value < 0.01; and ***, *p*-value < 0.0001.

## 3. Results

### 3.1. Sphingomyelin-, Cholesterol-, and GM1 Ganglioside-Enriched Domains Can Be Evidenced at the Myoblast Surface

To analyze SM, chol, and GM1 distribution at the outer PM leaflet, C2C12 myoblasts were single-labeled at 4 °C (to avoid probe endocytosis) with complementary probes, i.e., fluorescent lipid analogs (BODIPY-SM, TopFluor-Chol, and BODIPY-GM1) and/or toxin fragments/subunits (mCherry-lysenin, mCherry-theta, and Alexa 647-CTxB), and analyzed by vital high-resolution confocal microscopy [26,34,37,47]. Both X-Y basal sections and X-Z reconstitutions revealed that those lipids were not homogeneously organized but rather formed lipid clusters (Figure 1A). Those lipid clusters were similarly observed upon labeling at 20 °C and 37 °C instead of 4 °C (Appendix A). Moreover, sequential double-labeling of SM, chol or GM1 with toxin fragments/subunits followed by fluorescent lipid analogs revealed a high extent of colocalization between each pair of probes (Appendix A–C, insets). The quantification of the colocalization coefficients on the whole myoblast surface (i.e., not only domains) confirmed this qualitative observation (Appendix A–C, right), except for CTxB with BODIPY-GM1, which could be due to the stronger labeling intensity of the former (Appendix A, red CC). Those data indicated that, despite different sizes and physicochemical and spectral properties, each pair of probes was able to reveal the same lipid clusters.

To next exclude the possibility that these clusters could represent membrane protrusions, myoblasts were co-labeled with BODIPY-lipid analogs and the non-specific fluorescent membrane probe FM4-64X. A large dissociation between BODIPY-lipid clusters and FM4-64X-enriched areas was observed, suggesting that lipid clusters did not simply result from lipid enrichment at membrane ruffles (Figure 1B). Finally, to rule out that lipid labeling evidenced endocytic vesicles despite labeling at 4 °C, cells were labeled with BODIPY-SM or -GM1 and then treated with a high concentration of BSA to remove the probes from the PM. Most BODIPY-lipid labeling disappeared after treatment with BSA, indicating that the probes remained at the cell surface and were not internalized in endocytic vesicles (Figure 1C). 

To confirm the existence of submicrometric domains in a label-free manner with high spatial resolution while evaluating their membrane stiffness [33,36], AFM using lysenin- and theta-derived tips was conducted sequentially on the same cells. Results confirmed the presence of lipid clusters (Figure 1D, arrowheads). Those clusters presented a slightly lower membrane stiffness than the bulk membrane (Figure 1D, right), which agrees with the lower stiffness of lipid domains at the RBC surface [35,48]. A proportion of lipid clusters were revealed by both tips, suggesting co-enrichment in SM and chol (Figure 1D, yellow arrowheads). Altogether, these data indicated that SM-, chol-, and GM1-enriched domains were present at the myoblast surface and that these domains were softer than the bulk membrane.

### 3.2. Two Main Types of Domains, Enriched in SM/Chol/GM1 or in Chol, and Sensitive to Cholesterol Depletion, Coexist at the Myoblast Surface

To evaluate whether the above-mentioned domains coexisted at the myoblast surface or were instead redundant, we performed double lipid labeling. Data revealed a lower extent of colocalization for SM/chol and GM1/chol than for SM/GM1, the latter being close to the colocalization coefficient obtained upon double labeling with two different SM probes considered as the reference colocalization value (Appendix A). Those data suggested differential associations between lipids at the surface of myoblasts and that SM and GM1 were very well associated. Nevertheless, this analysis did not allow us to draw conclusions on lipid enrichment, specifically in domains, or on lipid domain proportion.

To address those questions, myoblasts were simultaneously triple-labeled for SM, chol and GM1, and their enrichment in domains was quantified using fluorescence intensity profiles drawn in myoblast periphery regions where domains were best visible (Figure 2A,B). Results showed that ~45% and ~25% of total domains at the myoblast surface were respectively enriched in SM/chol/GM1 and chol and that those two types of domains coexisted with a third, less abundant type of domain, enriched in SM/chol, and other domains even less abundant (Figure 2C). 

To further evaluate whether both main types of domains were effectively enriched in chol, the above double- and triple-labeling were reproduced upon treatment with mβCD to deplete chol. Treatment specificity and non-toxicity were first demonstrated by the three following experiments. First, mβCD specifically and reversibly reduced the membrane chol content proportional to drug concentration but did not affect the non-membrane esterified chol or the membrane SM contents (Figure 3A,B and Appendix A). Second, the drug specifically and reversibly decreased the surface labeling of chol but not those of SM and GM1, determined by the mean gray value (MGV) of the labeling (Figure 3C and Appendix A). Third, it slightly decreased the PM stiffness while preserving the cytoskeleton stiffness (Figure 3D,E). Data obtained upon double labeling indicated that chol depletion decreased SM/chol and GM1/chol colocalization but increased SM/GM1 colocalization (Appendix A), supporting the differential association between sphingolipids on the one hand and sphingolipids with chol on the other hand. Taking a closer look at the lipid domains, we found that chol depletion abrogated chol-enriched domains in a reversible manner, as expected (Figure 3F). It also strongly decreased the proportion of SM/chol/GM1- and chol-enriched domains and, to a lesser extent, the proportion of SM/chol-enriched domains in favor of SM-, SM/GM1-, and GM1-enriched domains without affecting the total number of lipid domains relative to the cell surface (Figure 3G–I). In conclusion, we evidenced the existence of two main types of submicrometric lipid domains at the surface of myoblasts and the possibility of modulating their proportion through chol depletion.

### 3.3. Chol- and SM/Chol-Enriched Domains Polarize at the Leading Edge and GM1-Enriched Domains at the Trailing Edge of Migrating Myoblasts

We next sought to evaluate if lipid domains were also relevant for myoblast migration. To do so, we started by SM, chol and GM1 single-labeling on migrating myoblasts in Ibidi chambers and visualization of lipid distribution by Airyscan confocal microscopy in super-resolution mode. We first verified that the labeling temperature did not affect myoblast morphology or the extent of migration (Appendix A). We then revealed the preferential distribution of SM and chol at the migration front, opposite to the distribution of GM1 at the trailing edge (Figure 4A and Appendix A). Quantification of lipid polarization by analysis of the fluorescence intensity at the front vs. the rear of the cell confirmed this polarized distribution (Figure 4B). We then switched to triple-labeling of migrating myoblasts and quantification of lipid domain proportion with fluorescence intensity profiles. This revealed that chol- and SM/chol-enriched domains were ~10-fold more abundant at the leading edge compared to the trailing edge, contrasting with 5-fold more GM1-enriched domains at the trailing edge compared to the leading edge (Figure 4C,E).

Three conclusions can be drawn from those experiments. First, the polarization of chol and SM at the leading edge vs. GM1 at the trailing edge was revealed by two independent labeling experiments and quantification procedures, i.e., single labeling followed by MGV analysis and triple labeling followed by fluorescence intensity profiles on domains. Second, a comparison of lipid domains on resting and migrating myoblasts indicated differential domain enrichment at the resting state and upon migration. Third, a comparison of domain proportion at resting and migrating states suggested a reorganization of lipid domains upon myoblast migration: SM/chol/GM1-enriched domains on myoblasts at resting state could dissociate to form oppositely polarized SM/chol- and GM1-enriched domains. In contrast, chol-enriched domains could persist in abundance and polarize at the front with SM/chol-enriched domain. This hypothesis was supported by the fact that the total number of lipid domains relative to the cell surface remained unchanged between resting and migrating myoblasts (Figure 4D).

### 3.4. Cholesterol Depletion Impairs Myoblast Migration and Sphingomyelin Polarization

Since the major domains present at the leading edge contained chol, we determined the impact of chol depletion on the extent of migration and the polarization of chol and SM. Two types of migration tests were developed: spontaneous migration in a cell-free surface of Ibidi chambers and oriented migration in Transwell chambers towards a gradient of IGF-1 [49]. Spontaneous and oriented migration decreased similarly after mβCD treatment (Figure 4F,G). Chol repletion with mβCD:chol complexes restored myoblast spontaneous migration, again supporting the non-toxicity of mβCD treatment (Figure 4F, triangle at 10 mM). Moreover, chol depletion decreased SM but not chol polarization (Figure 4H,I). Altogether, the above data suggested the implication of chol for myoblast movement, particularly the polarization of SM/chol-enriched domains at the migration front.

### 3.5. Sphingolipid Depletion Does Not Impair Chol-Enriched Domain Proportion at Resting State Neither Myoblast Migration nor Cholesterol and Sphingomyelin Polarization

To test whether the effects of chol depletion were specific or not, sphingolipid synthesis was inhibited by FB1. This drug was shown to decrease the SM content in a concentration-dependent manner without affecting the chol level (Appendix A), to specifically reduce the surface labeling of SM and GM1 but not of chol (Appendix A), to slightly increase the PM stiffness while preserving the cytoskeleton stiffness (Appendix A), and to impair SM- and GM1-enriched domains (Appendix A). The quantification of lipid domain proportion after triple-labeling indicated that FB1 decreased SM/chol/GM1-enriched domains to a similar extent as after mβCD treatment but preserved chol-enriched domains and increased those enriched in SM/chol (Figure 5A,C). This modification of lipid domain proportion was not due to a decrease in the total number of lipid domains relative to the cell surface (Figure 5B). We then used FB1 as a suitable tool to modulate lipid domains without affecting chol content to assess the effect of sphingolipid depletion on myoblast migration and lipid polarization. Results showed that neither spontaneous migration nor chol or SM polarization were affected. On the other hand, oriented migration seemed to slightly increase, although not significantly (Figure 5D–G). These results, combined with the fact that FB1 also increased SM/chol-enriched domain proportion at resting state (Figure 5C), strengthened our previous suggestion of the implication of chol- and SM/chol-enriched domain polarization for myoblast migration and confirmed the specific decreases in myoblast migration and SM polarization upon mβCD. 

### 3.6. Inhibition of Actin Polymerization Impairs Chol-Enriched Domain Proportion at Resting State, Myoblast Migration and Cholesterol and Sphingomyelin Polarization

We then evaluated whether chol- and SM/chol-enriched domains at the migration front could spatially associate with the actin cytoskeleton. To do so, migrating myoblasts were co-labeled with SiR-actin, a live-cell actin probe, and with lysenin or theta toxin fragment to simultaneously visualize filamentous actin (F-actin) and SM or chol upon myoblast migration. In agreement with the above data, SM and chol polarized at the migration front (Appendix A), and the proximity between SM- and chol-enriched domains and F-actin at the migration front was revealed by fluorescence intensity profiles (Appendix A).

To next test for a potential functional relationship between lipids, migration, and F-actin, we then disrupted the actin cytoskeleton using cytoD (Figure 6A) and evaluated its impact on myoblast migration and lipid polarization. CytoD treatment reversibly impaired myoblast spontaneous migration and, to a slighter extent, oriented migration (Figure 6B,C) and decreased both chol and SM polarization, although not significantly for SM (Figure 6D,E). This prompted us to analyze the chol distribution at the cell front in more detail. We found that, instead of being clustered in domains in the lamellipodia region, the chol distribution at the leading edge of cytoD-treated cells was less clustered and enriched in filopodia. This effect was specific since, upon mβCD, domains were smaller and less abundant but the lamellipodia area was still present (Appendix A). 

Finally, we asked whether the impairment of myoblast migration and chol polarization could be accompanied by an alteration of the lipid domain proportion at resting state. We found that cytoD did not affect either the membrane and total chol levels (Appendix A), or SM-, chol-, and GM1-enriched domains as shown by single-labeling and fluorescence intensity analysis for all three lipids (Figure 6F,G), nor total domain abundance (Figure 6I). Nevertheless, it altered lipid domain proportion with a strong decrease in chol-enriched domain abundance in favor of SM/chol/GM1-enriched domains that represented ~80% of total lipid domains present at the cell surface after cytoD treatment (Figure 6H,J). 

Thus, such as chol depletion, cytoD treatment decreased both myoblast migration and chol-enriched domain proportion as well as SM polarization; but, in contrast to chol depletion, it did not affect the chol content, increased the proportion of SM/chol/GM1-enriched domains, and decreased the polarization of chol in favor of chol-enriched filopodia all around the cell. Altogether, those data supported the relationship between the proportion of chol-enriched domains at the resting state and the extent of myoblast migration and showed the polarization of SM and chol in interplay with the cytoskeleton.

### 3.7. The Effects of Cholesterol Depletion on Chol-Enriched Domain Proportion at Resting State, Cholesterol and Sphingomyelin Polarization and Myoblast Migration Are Largely Abrogated in Actin-Depolymerized Myoblasts

To further test this interplay, myoblast migration, chol content, chol and SM polarization, and lipid domain abundance and proportion were evaluated upon treatment with mβCD, cytoD, or a combination of both compounds. To avoid cell toxicity potentially resulting from the combination of drugs, we used the lowest drug concentration to significantly impact lipid-enriched domains. As expected, the combined treatment decreased chol content similarly to mβCD treatment alone (Appendix A). More importantly, it induced similar effects as cytoD alone on both migration (Figure 7A,B), chol and SM polarization (Figure 7C,D), and chol-enriched domain proportion (Figure 7E–G), indicating the almost complete abrogation of the mβCD effect on those parameters in actin-depolymerized myoblasts. In contrast, the decreased proportion of SM/chol/GM1-enriched domains induced by mβCD was preserved in actin-depolymerized myoblasts (Figure 7F,H). Altogether, these data further supported that the proportion of chol- but not SM/chol/GM1-enriched domains at the resting state and chol and SM polarization at the front interplayed with the actin cytoskeleton.

### 3.8. Cholesterol Depletion Increases Focal Adhesion Surface Occupancy at the Cell Center Both in Resting and Migrating Myoblasts and Impairs Focal Adhesion and F-Actin Polarization at the Front and the Front Surface Size

To further study this hypothesis, we evaluated FA abundance and distribution upon mβCD combined or not with cytoD treatment on resting myoblasts through immunolabeling of paxillin, a FA scaffolding protein. Chol depletion tended to increase the total number of FAs relative to the cell surface and the surface occupancy, specifically in the center of the cell, whereas cytoD decreased both parameters (Figure 8A–C). The ratio of periphery vs. center FA surface occupancy was decreased whatever the treatment, and the effect of mβCD was abrogated in actin-depolymerized myoblasts (Figure 8D), suggesting the interplay between chol and F-actin in the control of FA distribution.

To confirm the effects of mβCD on FA distribution, resting and migrating cells were treated with increasing concentrations of mβCD, and FAs and F-actin parameters were measured. At both the resting state and upon migration, the FA total number tended to increase but not significantly (Appendix A and Figure 9A,B). Moreover, and in agreement with the data obtained above at 5 mM mβCD (see Figure 8C), the FA surface occupancy in the cell center also increased, but was more pronounced and proportional to the mβCD concentration (Appendix A), resulting in a proportional decrease of periphery vs. center FA surface occupancy (Appendix A). To better evaluate the impact of chol depletion on FA distribution at the different regions of migrating myoblasts, the proportion of FA surface occupancy in the front, center, and rear was calculated and compared to the total area of FAs. This analysis showed that FAs were polarized at the front in control conditions and that this polarization was progressively impaired with increasing concentrations of mβCD, first in favor of a higher central occupation, then in favor of both central and posterior occupation (Figure 9C). 

Chol depletion also slightly decreased the F-actin distribution at the cell front in favor of a higher central occupation without affecting the total cell surface occupancy (Figure 9D,E). Those modifications were accompanied by a mβCD concentration-dependent decrease of the cell front surface at the benefit of the cell center surface but with the preservation of the rear surface (Appendix A).

To ensure that these observations were the result of chol depletion and not of treatment toxicity and/or cell retraction, which could not be totally excluded at 10 mM mβCD (Appendix A), the experiment was repeated upon increasing concentrations of FB1. This treatment induced a decrease in the myoblast surface similar to mβCD treatment until 7.5 mM (Appendix A). Nevertheless, it affected neither the FA number and polarization nor the total cell surface F-actin occupancy and polarization (Figure 9F–I) nor the cell surface area of the front vs. the center or the rear (Appendix A). Those observations reinforced the hypothesis that chol depletion could specifically impair myoblast migration by altering the polarization of FAs and F-actin, as well as the front surface area.

### 3.9. The Lateral Diffusion of Domain-Associated Sphingomyelin at the Front Is Specifically Restricted in a Cholesterol- and Cytoskeleton-Dependent Manner

We finally tested whether SM/chol- and chol-enriched domains present at the leading edge could exhibit different biophysical properties than GM1-enriched domains at the trailing edge and whether this could depend on the chol and actin cytoskeleton. To do so, membrane lipid lateral diffusion was measured by FRAP on lipid domains vs. the bulk membrane area, both on resting and migrating myoblasts, while distinguishing the front and the rear. In resting myoblasts, the mobile fraction of SM and chol was ~50% vs. ~75% for GM1, suggesting that the lateral mobility of the former lipids was more restricted than the latter. Moreover, the lateral diffusion of SM associated with domains was significantly more restricted than SM in the bulk membrane, which was not the case for chol and GM1 (Figure 10A–C). Similar observations were made in myoblasts upon migration, revealing SM-specific higher restriction in domains than in the bulk membrane at the front and higher restriction at the front than at the rear (Figure 10D–F). This restriction was entirely abrogated upon combined mβCD and cytoD treatments, suggesting that the SM-restricted mobility at the front was related to both chol and the actin cytoskeleton. In contrast, chol and GM1 membrane lateral mobility was not affected by those treatments (Figure 10G–I). Altogether, those data supported the specific functional interplay between SM-enriched domains, the F-actin network, and chol at the leading edge. 

## 4. Discussion

### 4.1. Main Observations

We showed here the coexistence of two main types of submicrometric domains, respectively enriched in SM/chol/GM1 and in chol, at the outer PM leaflet of resting myoblasts. We also evidenced the polarization of chol and SM at the leading edge and a more complex lipid domain distribution in migrating myoblasts. Indeed, we found three main types of domains: those enriched in chol and in SM/chol at the leading edge, and those enriched in GM1 at the trailing edge. In addition to differences in lipid composition and polarized distribution, those domains differ in terms of cytoskeleton interplay and biophysical properties. Our data further suggest that domains at the leading edge could help in lamellipodia formation/maintenance, and FA assembly/disassembly.

### 4.2. Experimental Strategy Strengths and Weaknesses

Complementary lipid tools were used in this study, i.e., fluorescent lipid analogs [26,50,51,52,53] and toxin fragments specific to endogenous lipids [26,34,37]. Labeling with these tools separately gave rise to similar domains. Moreover, excellent colocalizations in domains and colocalization coefficients between ~0.5 and 0.8 (except for CTxB with BODIPY-GM1) were obtained upon sequential labeling with two probes for the same lipid. 

Regarding the imaging methods, although confocal microscopy offers several advantages, the size of the observed lipid domains could have been overestimated. This is the reason for which we also used Airyscan confocal microscopy in super-resolution mode, which improves resolution in X-Y by ~2.5-fold as compared to confocal microscopy. We also confirmed the existence of submicrometric domains in a label-free manner with AFM using lysenin- and theta-derived tips to benefit from still higher resolution than Airyscan while evaluating domain membrane stiffness [33,36]. 

Three complementary methods were developed for the quantification of lipid distribution on the acquired images. First, the MGV used to measure the global fluorescence intensity on entire cells or on specific areas allowed to evaluate the global enrichment of lipids between different conditions and/or different areas of the cell. Second, the use of colocalization coefficients allowed for the determination of the extent of lipid colocalization. However, these methods were not specific to lipid domains and took the bulk membrane into account. To counteract this drawback, we developed a third method based on fluorescence intensity profiles to specifically evaluate lipid domain enrichment, allowing us to generate a mapping of the different domains at the myoblast surface. 

To modulate lipid domains, four complementary and validated approaches (i.e., chol depletion, sphingolipid synthesis inhibition, and actin polymerization inhibition, combined or not with chol depletion) were compared for their effect on lipid polarization, lipid domain abundance and proportion, and myoblast migration. Although we cannot conclude that the effects of chol depletion on myoblast migration were only attributed to the chol-enriched domains, we could be confident that chol-enriched domains represented a major target of mβCD, as supported by the following lines of evidence. First, specificity of mβCD towards the PM and not the cytoskeleton was proved by AFM since the cytocortex stiffness was fully preserved in contrast to the bulk PM stiffness which was slightly decreased at 5 mM mβCD. Second, the very slight decrease of the PM stiffness contrasted with the stronger reduction of the proportion of chol-enriched domains and could suggest that chol depletion mainly affected domains and less the bulk membrane. Third, due to the high threshold required for chol theta toxin binding (∼30mol%; [54,55]), domains should present a very high chol content. Accordingly, they were decreased by ~50% at 5 mM mβCD, in the same order of magnitude as the ~50% reduction of total chol content. 

### 4.3. Evidence for Two Main Types of Submicrometric Lipid Domains at the Surface of Resting Myoblasts

It is now accepted that the PM organization is far more complex than previously thought [26]. Lipid rafts were first described as platforms enriched in chol, sphingolipids, and PM proteins involved in the regulation of cell function [56]. More recently, submicrometric domains with differential lipid composition were evidenced at the surface of several types of living cells [26,48], but we are far from a generalization of the concept. We showed here the presence of two main types of submicrometric domains at the outer PM leaflet of myoblasts, enriched in SM/chol/GM1 or in chol. To the best of our knowledge, this is the first detailed description of such a heterogeneous lipid distribution for this cell type. Those domains together represented ~70% of the total domains present on resting myoblasts, exhibited a lower membrane stiffness compared to the bulk membrane, and were both sensitive to chol depletion. 

Despite their similarities, several lines of evidence suggested that these two types of domains were not redundant but coexisted at the resting myoblast surface. First, their proportion was quite different, representing ~45% (SM/chol/GM1) and 25% (chol) of total domains. Second, their proportion was oppositely modified by inhibition of actin polymerization, and only the former domains were decreased by sphingolipid depletion. Third, whereas the proportion of chol-enriched domains was similar in myoblasts at resting state and upon migration, the proportion of SM/chol/GM1-enriched domains was drastically reduced. These observations, and hence the generalization of lipid domain coexistence, are supported by our demonstration that three types of submicrometric lipid domains coexist at the surface of living RBCs, respectively enriched in chol, SM/chol, and GM1/chol [34,35,37].

### 4.4. Evidence for Cholesterol and Sphingolipid Polarization and Their Clustering in Three Main Types of Submicrometric Domains upon Myoblast Migration

During myoblast migration, the PM is extremely solicited and has to present the appropriate biophysical properties, such as tension or fluidity, at the adequate moment [18,19,25]. Since chol is a key regulator of those properties [57,58], it is not surprising that chol depletion impaired myoblast spontaneous and oriented migration. The role of chol in cell migration has been shown in other cell types. For example, changes in chol levels in sarcoma cell lines have a specific effect on signaling pathways involved in cell migration [59]. Similarly, the increase in chol levels was shown to inhibit liver cancer cell migration and invasion, while depletion of chol compromises breast cancer cell migration [60,61]. Lastly, increased PM chol content inhibits macrophage migration [62]. These studies mainly focused on chol content and not on chol (re)organization at the PM. We showed here that chol was important for myoblast migration through its clustering in domains and not its content, as chol content did not correlate with spontaneous myoblast migration and very slightly with oriented migration (Appendix A).

Among the four types of chol-containing domains detected at the PM of resting myoblasts (i.e., SM/chol/GM1, chol, SM/chol, and GM1/chol), only those enriched in chol and SM/chol appeared to play a role at the leading edge of migrating myoblasts (Figure 11J), as supported by the following evidences. First, chol- and SM/chol-enriched domains were far more abundant than SM/chol/GM1- and GM1/chol-enriched domains at the leading edge, representing respectively ~20, 25, 10, and 8% of total domains present at the front. This was confirmed by the quantification of lipid polarization, showing a ~2-fold enrichment in chol and SM at the front vs. ~0.8 in GM1. Chol- and SM/chol-enriched domain proportions at resting state and chol and SM polarizations very well correlated (except for SM/chol-enriched domain proportion and chol polarization; Figure 11B,C,F,G), supporting the relation between lipid domains at resting state and lipid polarization. Second, the comparison of the effects of chol depletion, sphingolipid synthesis inhibition, and chol depletion combined with F-actin depolymerization on lipid domain proportion and myoblast migration revealed that both chol- and SM/chol-enriched domain proportion were very well correlated with both spontaneous and oriented migration (Figure 11A,E). In contrast, the SM/chol/GM1-enriched domain proportion did not correlate at all with the extent of myoblast migration (Appendix A), and GM1/chol-enriched domains were affected by none of the pharmacological treatments.

On the contrary to chol- and SM/chol-enriched domains, those enriched in GM1 were by far the domains most present at the trailing edge (Figure 11J). In terms of relative proportion, SM/chol- and GM1-enriched domains represented together ~45% of total domains present at the myoblast surface upon migration, which exactly corresponded to the proportion of SM/chol/GM1-enriched domains at the resting myoblast surface. Since the total abundance of lipid domains relative to the cell surface did not change between resting and migrating cells, this suggested that the preexisting SM/chol/GM1-enriched domains were separated and redistributed upon myoblast migration. 

Those data illustrated the relevance of lipid domain reorganization for physiological cell migration. In agreement with our findings that SM/chol-enriched domains polarize at the leading edge, the implication of SM-enriched domains in migration has been shown in keratinocytes [28]. However, opposite results can be found in T lymphocytes and mouse embryonic fibroblasts, where the breakdown of SM at the leading edge has been shown to favor directional cell migration [63,64]. Regarding GM1, it has been shown to polarize at the rear of T lymphocytes [30] or at the leading edge of epithelial ECV304 cells, in which it seems to be involved in directional selection for cell migration [29]. Differences in lipid polarization could be partially explained by differential migration modes, as epithelial cells migrate in a collective way opposite to the single-cell mesenchymal myoblast migration [65,66]. Even though it is important to take into account that modification of cell confinement and the underlying substrate can induce collective migration, myoblasts preferentially migrate in a single-cell mode in vitro [6,7,8,11,67]; our experimental settings with respect to cell confluency indeed favored this mode.

### 4.5. Chol- and SM/Chol-Enriched Domains at the Leading Edge Present Differential Interplay with the Cytoskeleton and Biophysical Properties

On resting myoblasts, domains enriched in chol and SM appeared to exhibit similar biophysical properties and actin cytoskeleton dependence. Indeed, AFM revealed a similar, slightly lower stiffness of SM- and chol-enriched domains as compared to non-adhesive areas. Moreover, FRAP experiments showed similar ~50% mobile fractions of SM and chol after domain photobleaching, suggesting similar restriction to their lateral mobility. In addition, the proportion of chol- and SM/chol-enriched domains was similarly affected by mβCD and cytoD. Likewise, on migrating myoblasts, SM and chol polarization were similarly decreased by cytoD, but their lateral mobility in the bulk membrane was not affected, suggesting that the actin cytoskeleton was similarly required for appropriate polarization of those lipids but not for their lateral diffusion in the membrane. 

In contrast to cytoD, chol depletion by mβCD differentially affected both polarization at the leading edge and lateral diffusion of SM vs. chol. For instance, SM polarization was decreased by mβCD and its lateral mobility in the bulk membrane at the leading edge, which was more restricted than at the trailing edge, was abrogated by the drug. This was not the case for chol, suggesting that SM, and not chol, depended on the membrane chol content for its polarization and lateral mobility. Moreover, in addition to differences between the bulk membrane at the leading and trailing edges, SM lateral diffusion associated with domains was more restricted than in the bulk. Since this restriction was fully abrogated by combined mβCD and cytoD treatments but not by drugs alone, this suggested that the SM restriction to lateral mobility at the front was dependent on both chol and the actin cytoskeleton. This was again not the case for chol. 

In conclusion, through the comparison of domains at resting state and lipid and domain polarization, we concluded that chol and SM can be distinguished based on both interplay with the cytoskeleton and biophysical properties only in domains and at the leading edge, supporting the possibility of differential roles for those domains, as discussed in the next section.

### 4.6. Chol- and SM/Chol-Enriched Domains at the Leading Edge Present Differential Roles in Myoblast Migration

The implication of chol- and SM/chol-enriched domains at the leading edge is supported by the excellent positive correlations of their abundance with myoblast migration, chol and SM polarization, and FA distribution at the cell periphery, except for SM/chol domains with chol polarization (Figure 11A–H). Several arguments supported the possibility that these two types of domains contributed to myoblast migration by different mechanisms. First, although the proportion of SM/chol-enriched domains was minor compared to chol-enriched domains on resting myoblasts (~5% vs. ~25%), they became even more abundant than chol-enriched domains upon migration at the expense of SM/chol/GM1-enriched domains, which suggests a role for those domains as well. Second, both domains presented a differential interplay with the cytoskeleton and different biophysical properties, as discussed in Section 4.5. Third, upon FB1, the increase of SM/chol-enriched domains was accompanied by a slight increase in migration, whereas chol-enriched domains were preserved. Fourth, the coexistence and differential roles of chol- and SM/chol-enriched domains at the leading edge vs. GM1 at the trailing edge were reminiscent of the coexistence of three populations of lipid domains at the RBC surface and their different roles during RBC deformation [35].

Based on all our observations, we hypothesize that the two types of domains at the leading edge contribute to myoblast migration by distinct mechanisms. The first mechanism implies the contribution of chol-enriched domains in the formation/maintenance of lamellipodia. In support of this hypothesis, we showed that chol depletion decreased the size and abundance of chol-enriched clusters in lamellipodia as well as the front cell surface size, to the benefit of the cell center. On the other hand, actin polymerization inhibition induced the formation of chol-enriched filopodia all around the cell. On a mechanistic point-of-view, chol-enriched domains could provide the optimal membrane stiffness to support the mechanical forces driven by actin polymerization and/or optimal membrane curvature to allow for PM deformation during lamellipodia formation. The membrane stiffness hypothesis is supported by a previous study that highlighted in fish keratocytes a gradient of rigidity during migration [19]. Likewise, on neuronal cells, chol depletion was shown to decrease membrane stiffness in the lamellipodium, which is an important factor for force generation [68]. Nevertheless, the lack of difference between chol lateral diffusion at the leading vs. the trailing edge observed here did not support this hypothesis. On the other hand, the membrane curvature hypothesis is in agreement with the increase in vesicle deformation by F-actin protrusions upon rise in chol levels, as revealed on actin-encapsulated lipid vesicles [69], and with the gathering of chol-enriched domains in increased curvature areas upon RBC deformation [52].

The second mechanism involves the recruitment of proteins involved in FA assembly/disassembly by SM/chol-enriched domains. The role of chol in this process is not new and is supported by studies in different cell types showing the loss of periphery FAs and the relocation of paxillin towards the center of the cell as well as the decrease in focal adhesion kinase (FAK) activity and in membrane stiffness upon chol depletion [59,70,71]. One step further, we here propose the role of SM/chol-enriched domains in this process, based on the specific restriction to SM lateral mobility at the leading edge, its dependence on chol and F-actin (see Section 4.5), and the positive correlation between chol- or SM/chol-enriched domains with FA distribution combined with the inverse correlation between domain-associated SM lateral mobility and cell migration (Figure 11D,H,I). Two non-mutually exclusive hypothetical mechanisms can be proposed. In the first one, domains could sequester activated integrins by providing a favorable membrane environment in terms of lipid packing and/or thickness. In support of this hypothesis, changes in chol levels have been shown to alter integrin sequestration in raft-mimicking lipid mixtures [72]. The second hypothesis implies the role of domains in integrin-mediated signal transduction pathways initiated by cell adhesion. This possibility was supported by the observations that (i) changes in PM chol levels modulate cell signaling and regulate cell adhesion and migration on fibronectin [60]; (ii) the oxysterol-binding protein-related protein 2 (ORP2) couples LDL-chol transport to FAK activation by endosomal chol/PI4,5P_2_ exchange [73]; and (iii) lipid rafts appear to contribute to integrin-mediated signaling [74]. Nevertheless, we cannot exclude the possibility that the role of SM/chol-enriched domains was indirect, involving PI4,5P_2_ clusters at the inner leaflet, previously shown to superpose with SM domains at the outer PM leaflet [75]. These PI4,5P_2_ clusters are indeed enriched at FA sites and interact with several FA proteins, including talin, vinculin, and FAK [74]. The interaction of talin with the cell membrane is in turn essential for integrin activation and FA formation [76].

### 4.7. Conclusions

Our data underlined that chol and SM polarization at the leading edge of migrating myoblasts contributes to their migration. Their clustering in submicrometric domains could spatially and functionally control myoblast migration by interacting with the cytoskeleton and/or providing the appropriate membrane biophysical environment for cell polarity establishment.

## Figures and Tables

**Figure 1 biomolecules-13-00319-f001:**
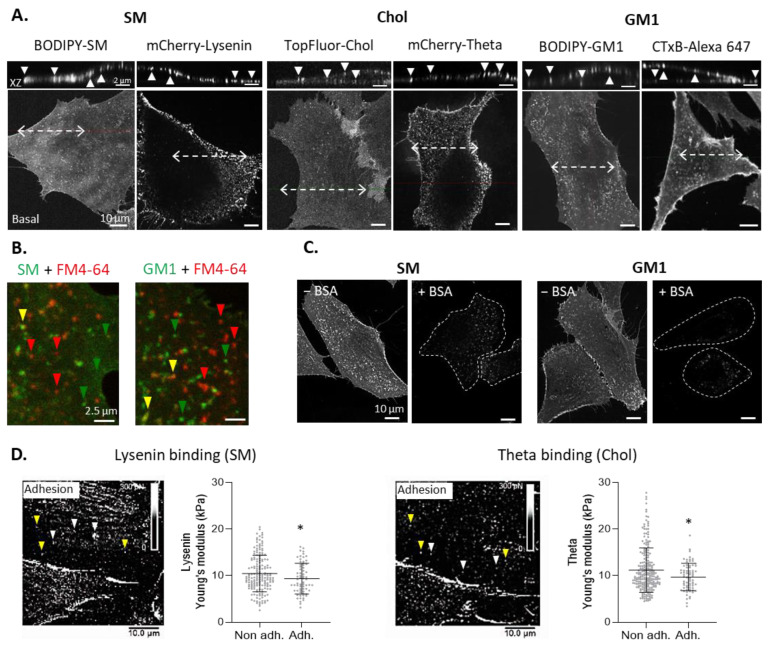
Clusters enriched in sphingomyelin, cholesterol, or GM1 ganglioside are revealed at the outer plasma membrane of C2C12 myoblasts and exhibit lower stiffness than the bulk membrane. (**A**) Basal confocal sections and orthogonal XZ reconstructions of cells single-labeled for SM (BODIPY-SM or mCherry-lysenin), chol (TopFluor-Chol or mCherry-theta), or GM1 (BODIPY-GM1 or Alexa 647-CTxB). Dotted arrows are cell sections for XZ reconstructions. Arrowheads are lipid clusters. Representative of >5 independent experiments for each probe except TopFluor-Chol (*n* = 2). (**B**) Images of cells double-labeled for SM (BODIPY-SM) or GM1 (BODIPY-GM1) and the non-specific membrane probe FM4-64X. Green and red arrowheads are BODIPY-lipid and FM4-64X enrichments; yellow arrowheads indicate colocalization. One experiment. (**C**) Images of cells labeled for SM or GM1 (BODIPY-SM or GM1), then directly imaged (left), or treated with 5% BSA (right). Dotted lines are cell outlines. One experiment. (**D**) Cells analyzed by AFM with lysenin- and theta-coupled tips to specifically bind SM or chol. Binding events with a force >100 pN represent adhesive areas (adh.), whereas a force <30 pN represents non-adhesive areas (non adh; adhesion maps, right). White arrowheads have an adhesive area for one lipid; yellow arrowheads have an adhesive area for SM and chol. Young’s modulus was extracted from the information gathered by the AFM tip in adhesive and non-adhesive areas (graphs, right). The data are expressed as means ± SD (*n* = 6 cells from 1 experiment with 70 measures/condition). Unpaired *t*-test. The statistics above for adhesive area groups refer to non-adhesive area groups.

**Figure 2 biomolecules-13-00319-f002:**
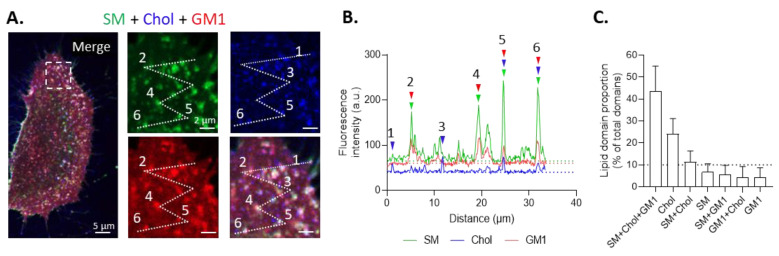
Two main types of domains, enriched in sphingomyelin/cholesterol/GM1 ganglioside vs. cholesterol, mainly coexist at the outer plasma membrane of C2C12 myoblasts. (**A**,**B**) Cells were triple-labeled for SM (BODIPY-SM), chol (theta), and GM1 (CTxB), visualized by confocal microscopy, and analyzed for lipid domains at the periphery basal side by drawing profiles (dotted white line) and generating graphs of the fluorescence intensity vs. the profile distance. The profiles were manually drawn and designed to cross as many lipid domains as possible in the entire cell periphery. Arrowheads are the domains enriched in lipids from corresponding colors. Green, blue, and red dotted lines indicate the threshold for each fluorescent channel. (**C**) The quantification of lipid domain proportion. Dotted line is the threshold for lipid domains acknowledgement. The data are expressed as means ± SD (*n* = 60 cells from 11 independent experiments, 10–15 profiles drawn, and 150–200 domains analyzed/cell).

**Figure 3 biomolecules-13-00319-f003:**
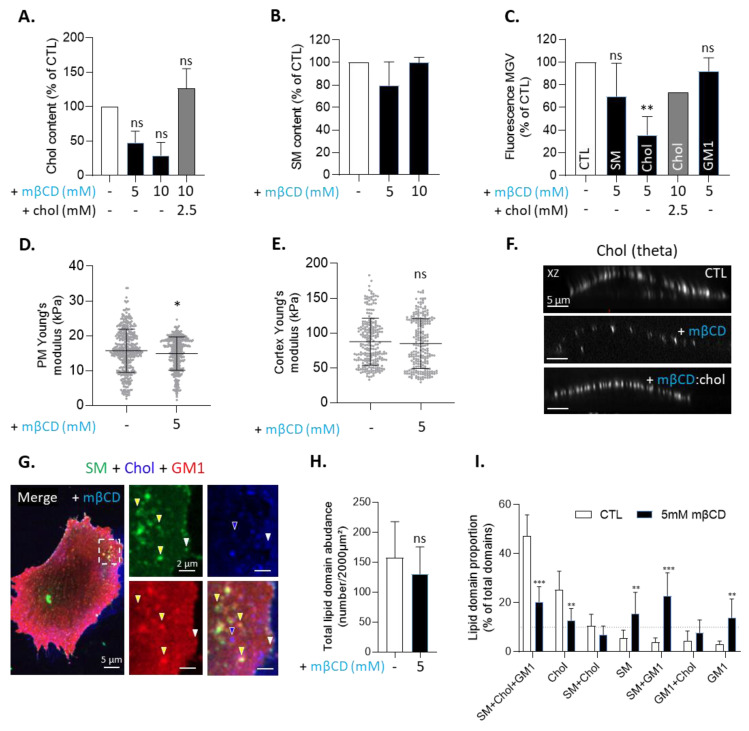
Specific and reversible membrane cholesterol removal by methyl-β-cyclodextrin similarly decreases the proportion of SM/chol/GM1- and chol-enriched domains. (**A**–**C**) C2C12 myoblasts were treated with the indicated concentrations of mβCD to deplete chol, followed or not by chol repletion. Cells were then assessed for chol (*n* = 3) independent experiments; (**A**) or SM content (*n* = 2 independent experiments; (**B**) or single-labeled for SM, chol, or GM1 (lysenin, theta, CTxB) to quantify basal fluorescence MGV (*n* = 1–3 independent experiments; 6–13 cells analyzed per condition; (**C**). The data are expressed as % of control (means ± SD). Kruskal–Wallis test, followed by Dunn’s multiple comparisons test. (**D**,**E**) Plasma membrane (left) or cortex (right) rigidity of myoblasts left untreated or treated with mβCD evaluated by AFM. The data are expressed as means ± SD (*n* = 6 cells from 1 experiment with 50 measures/cell). Unpaired *t*-test. (**F**) Cells were treated or not with 5 mM mβCD, followed or not by chol repletion, then labeled for theta and visualized by confocal microscopy with XZ reconstructions. Representative of three independent experiments. (**G**–**I**) Cells were pretreated or not with mβCD, then triple-labeled (BODIPY-SM, theta, and CTxB) and quantified for total lipid domain abundance relative to cell surface (**H**) and lipid domain proportion (**I**) as in Figure 2. White arrowheads are SM/chol/GM1-enriched domains. Blue arrowheads are chol-enriched domains. Yellow arrowheads are SM/GM1-enriched domains. Dotted line is the threshold for lipid domain acknowledgement. The data are expressed as mean ± SD (*n* = 10–12 cells for each condition from 2 independent experiments; 10–15 profiles drawn and 150–200 domains analyzed/cell). Wilcoxon signed-ranked test and two-way ANOVA were followed by Sidak’s multiple comparisons test. The statistics above the columns refer to the corresponding control.

**Figure 4 biomolecules-13-00319-f004:**
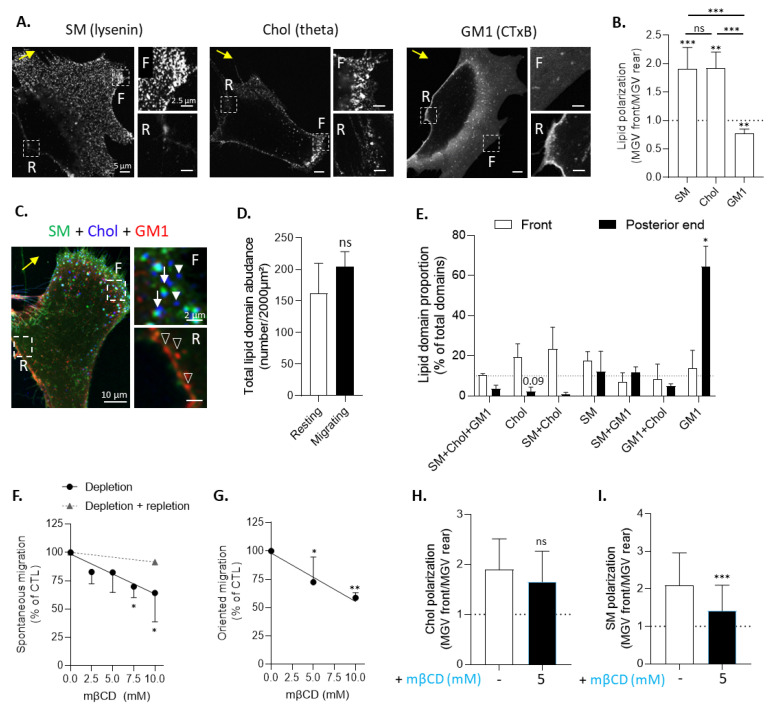
Myoblast migration implies the polarization of chol- and SM/chol-enriched domains at the leading edge vs. GM1-enriched domains at the trailing edge and is impaired by cholesterol depletion. (**A**) Spontaneous migrating cells in Ibidi chambers are single-labeled for SM (lysenin), chol (theta), or GM1 (CTxB) and visualized by super-resolution Airyscan confocal microscopy. Yellow arrow indicates the direction of migration; F is the front; and R is the rear. (**B**) The quantification of lipid polarization. The data are expressed as the ratio of the MGV at the front vs. the MGV at the rear (*n* = 10–11 independent experiments). Dotted line is the control value for no polarization. Kruskal–Wallis test, followed by Dunn’s multiple comparisons test. (**C**) Migrating cells were triple-labeled for SM (BODIPY-SM), chol (theta), and GM1 (CTxB) and visualized by super-resolution Airyscan confocal microscopy. Yellow arrow indicates the direction of migration; empty arrowheads are the GM1-enriched domains; white arrows are the SM/chol-enriched domains; and white arrowheads are the chol-enriched domains. (**D**,**E**) The quantification of lipid domain abundance relative to the cell surface (**D**) and lipid domain proportion at the front and rear of migrating cells (**E**). Dotted line is the threshold for lipid domain acknowledgement. The data are expressed as means ± SD (*n* = 3 cells from 1 experiment, 10–15 profiles/zone).Wilcoxon signed-ranked test and two-way ANOVA followed by Sidak’s multiple comparisons test. (**F**,**G**) The quantification of myoblast spontaneous (**F**) or oriented (**G**) migration in Ibidi chambers or transwells towards IGF-1 after mβCD treatment followed or not by chol repletion (triangle and dotted line). The data are expressed as mean ± SD (*n* = 4–11). Kruskal–Wallis test, followed by Dunn’s multiple comparisons test. (**H**,**I**) The quantification on single-labeled cells of chol (theta, **H**) and SM (lysenin, **I**) polarization after 5 h of migration in control conditions or after pretreatment with mβCD. The data are expressed as the ratio of the MGV at the front vs. the MGV at the rear (*n* = 45–53 cells for each condition from 6 independent experiments). Dotted line indicates no polarization. Unpaired *t*-test. The statistics above the columns refer to the corresponding control while statistics between different groups are indicated with bars on top of the graphs.

**Figure 5 biomolecules-13-00319-f005:**
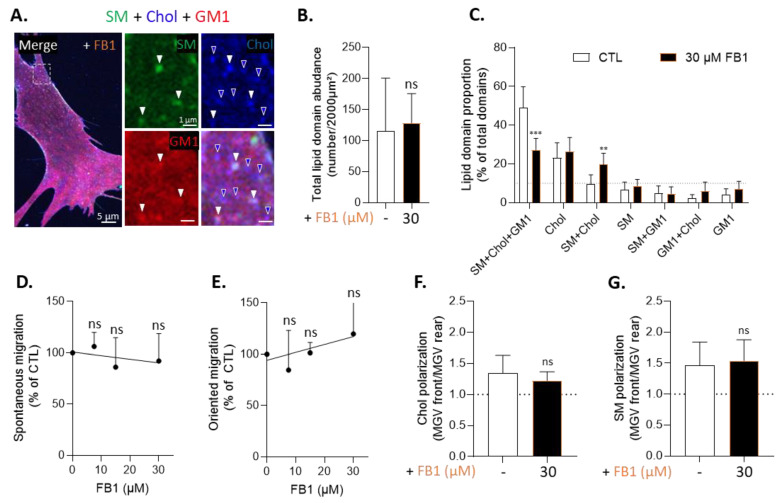
Sphingolipid synthesis inhibition by fumonisin B1 decreases the proportion of SM/chol/GM1- but not of chol-enriched domains and does not decrease myoblast migration nor lipid polarization. (**A**–**C**) Cells were pretreated or not with FB1 then triple-labeled (BODIPY-SM, theta, CTxB; (**A**) and quantified for lipid domain abundance relative to the cell surface (**B**) and lipid domain proportion (**C**), as in Figure 2. White arrowheads indicate the SM/chol/GM1-enriched domains; blue arrowheads indicate the chol-enriched domains. Dotted line indicates the threshold for lipid domain acknowledgement. The data are expressed as mean ± SD (*n* = 10 cells for each condition from 2 independent experiments; 10–15 profiles drawn and 150–200 domains analyzed/cell). Wilcoxon signed-ranked test and two-way ANOVA followed by Sidak’s multiple comparisons test. (**D**,**E**) The quantification of myoblast spontaneous (**D**) or oriented (**E**) migration in Ibidi chambers or transwells towards IGF-1 after FB1 treatment. The data are expressed as mean ± SD (*n* = 5–10 independent experiments). Kruskal–Wallis test followed by Dunn’s multiple comparisons test. (**F**,**G**) The quantification on single-labeled cells of chol (theta, **F**) and SM (lysenin, **G**) polarization after 5 h of migration in control conditions or after pretreatment with FB1. The data are expressed as the ratio of the MGV at the front vs. the MGV at the rear (*n* = 8–16 cells from 1–2 independent experiments). Dotted line indicates no polarization. Unpaired *t*-test. The statistics above the columns refer to the corresponding control.

**Figure 6 biomolecules-13-00319-f006:**
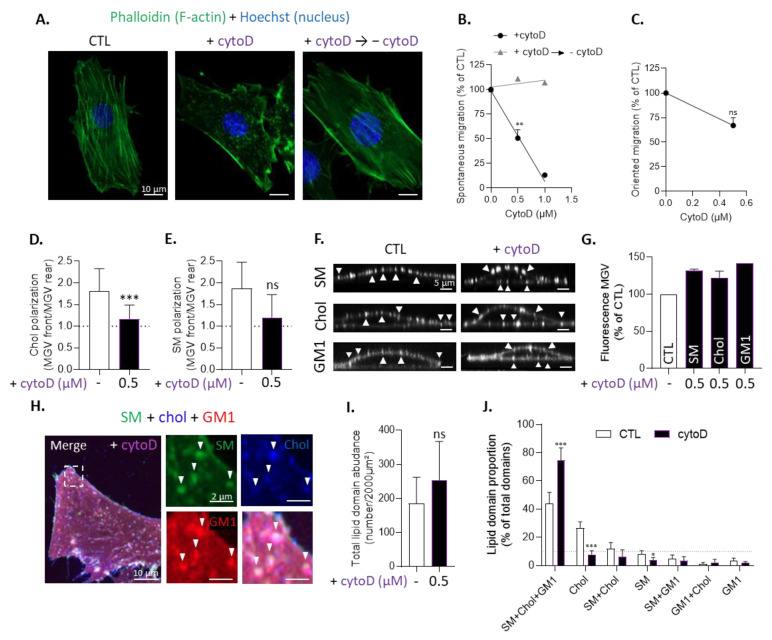
Actin polymerization inhibition by cytochalasin D decreases myoblast migration, cholesterol and sphingomyelin polarization, and chol-enriched domain proportion without depleting membrane lipids. (**A**) Cells were either left untreated (CTL) or pretreated with cytochalasin D (cytoD), then incubated for 5 h in medium still containing cytoD (+cytoD) or in cytoD-free medium (+cytoD -> −cytoD). Cells were then labeled for nucleus (blue) and F-actin (green). (**B**,**C**) Cells were treated the same way as in A with increasing concentrations of cytoD then migrated in Ibidi chambers (**B**) or transwells towards IGF-1 (**C**). The data are expressed as means ± SD (*n* = 1–5 independent experiments). Wilcoxon signed-rank test. (**D**,**E**) The quantification on single-labeled cells of chol (theta, D) and SM (lysenin, **E**) polarization after 5 h of migration in control conditions or after pretreatment with cytoD. The data are expressed as the ratio of the MGV at the front vs. the MGV at the rear (*n* = 30–35 cells from 4 independent experiments). Dotted line indicates no polarization. Unpaired *t*-test. (**F**,**G**) Control or cytoD pretreated cells were single-labeled for SM (BODIPY-SM), chol (theta), or GM1 (CTxB), then visualized (**F**) and analyzed for their basal fluorescence MGV (**G**). The data are expressed as means ± SD (*n* = 1–2 independent experiments). Arrowheads indicate the lipid domains. (**H**–**J**) Control or cytoD pretreated cells were triple-labeled for SM, chol and GM1 (BODIPY-SM, theta, CTxB), then quantified for lipid domain abundance relative to cell surface (**I**) or proportion (**J**), as in Figure 2. White arrowheads indicate the SM/chol/GM1-enriched domains. Dotted line indicates the threshold for lipid domain acknowledgement. The data are expressed as means ± SD (*n* = 6 cells from 1 experiment, 10–15 profiles drawn and 150–200 domains analyzed/cell). Wilcoxon signed-rank test or two-way ANOVA followed by Sidak’s multiple comparisons test. The statistics above the columns refer to the corresponding control.

**Figure 7 biomolecules-13-00319-f007:**
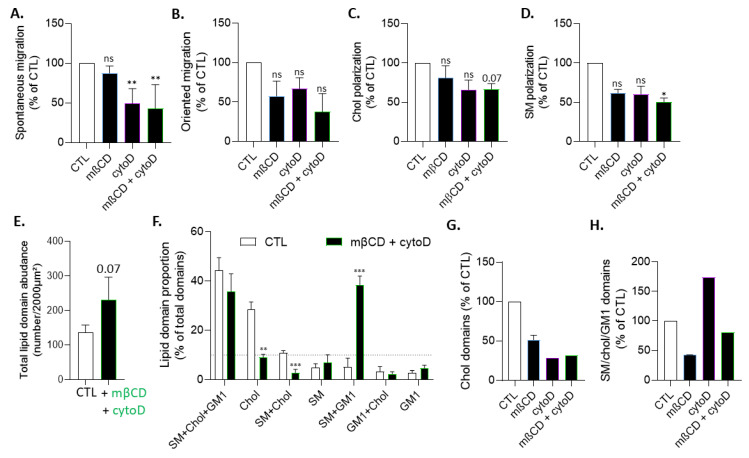
The effects of cholesterol depletion on chol-enriched domain proportion at resting state, cholesterol and sphingomyelin polarization, and migration extent are largely abrogated in actin-depolymerized myoblasts. Cells were left untreated or pretreated with 5 mM mβCD, 0.5 µM cytoD or a combination of both treatments. (**A**,**B**) The quantification of cell migration in Ibidi chambers (**A**) or in transwells towards IGF-1 (**B**). The data are expressed as means ± SD (*n* = 3–7 independent experiments). Friedman test followed by Dunn’s multiple comparisons test. (**C**,**D**) Cells were single-labeled for chol (theta) or SM (lysenin) then analyzed for lipid polarization. The data are expressed as the MGV at the front vs. the MGV at the rear then in % of control (means ± SD, *n* = 3 independent experiments). Friedman test followed by Dunn’s multiple comparisons test. (**E**,**F**) Cells were simultaneously triple-labeled for SM, chol, and GM1 (BODIPY-SM, theta, CTxB), then quantified for lipid domain abundance relative to cell surface (**E**) and proportion (**F**), as in Figure 2. Dotted line indicates the threshold for lipid domain acknowledgement. The data are expressed as means ± SD (*n* = 4–6 cells from 1 experiment, 10–15 profiles drawn and 150–200 domains analyzed). Wilcoxon signed-rank test and two-way ANOVA followed by Sidak’s multiple comparisons test. (**G**,**H**) Proportion of chol- and SM/chol/GM1-enriched domains abundance from respectively Figure 3I, Figure 6J, and Figure 7F to compare the effects of treatments expressed in % of internal control (means ± SD, *n* = 1–2 independent experiments). The statistics above the columns refer to the corresponding control.

**Figure 8 biomolecules-13-00319-f008:**
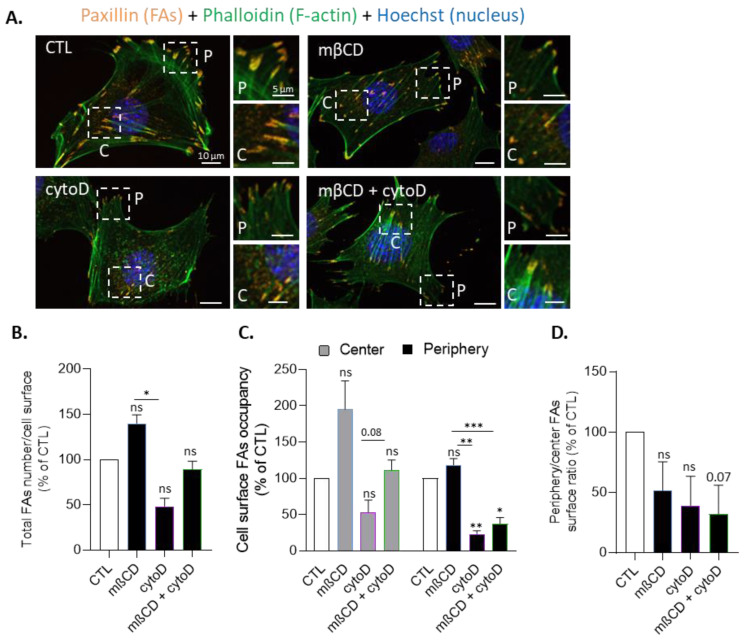
Cholesterol depletion and actin polymerization inhibition oppositely affect the total focal adhesion number and cell surface occupancy both at the center and the periphery of resting myoblasts but similarly impair their distribution at the periphery vs. the center. Cells were pretreated with 5 mM mβCD, 0.5 µM cytoD, or a combination of both treatments then (immuno)labeled for F-actin (green), paxillin (orange), and nuclei (blue). (**A**) Representative images. P, periphery; C, center. (**B**–**D**) The quantification of FA number/cell surface (**B**), FA cell surface occupancy in the center, and periphery of the cell (**C**), and corresponding periphery/center FAs surface ratio (**D**). The data are expressed in % of the control (means ± SD, *n* = 3 independent experiments). Friedman test followed by Dunn’s multiple comparisons test and two-way ANOVA. The statistics above the columns refer to the control while statistics between different treatments are indicated with bars on top of the graphs.

**Figure 9 biomolecules-13-00319-f009:**
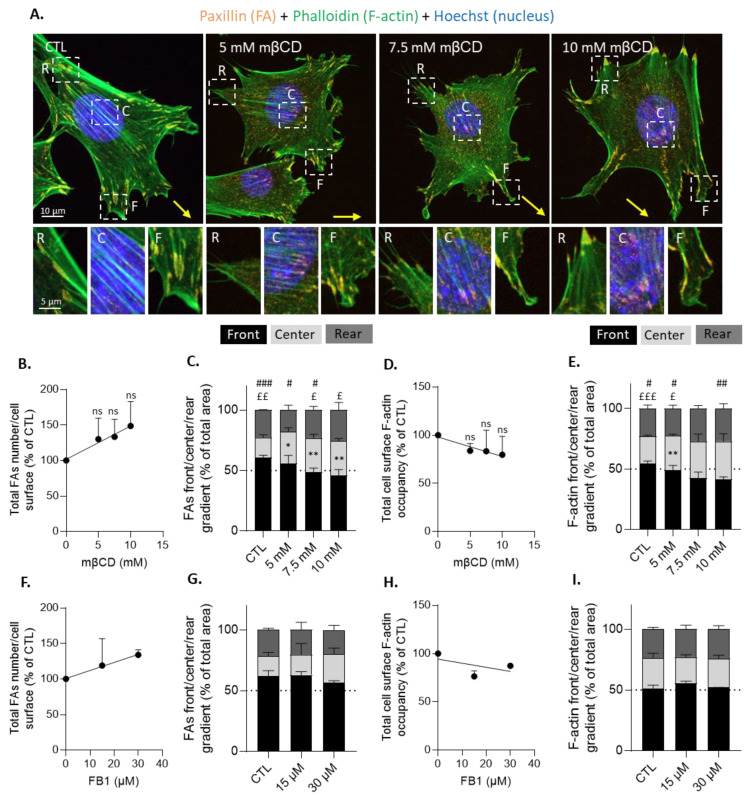
In contrast to sphingolipid synthesis inhibition, cholesterol depletion impairs polarization of focal adhesions and to a lower extent of F-actin upon myoblast migration. Cells were pretreated with the indicated concentrations of mβCD (**A**–**E**) or FB1 (**F**–**I**), migrated for 5 h in Ibidi chambers, and then immunolabeled for F-actin (green), paxillin (orange), and nuclei (blue). (**A**) Representative images. R, rear; C, center; F, front. Arrows indicate the direction of migration. (**B**–**I**) The quantification of FA total number relative to cell surface (**B**,**F**), FA polarization (**C**,**G**), F-actin total cell surface occupancy (**D**,**H**), and F-actin polarization (**E**,**I**). The polarization data are expressed as % of FAs or F-actin total area (means ± SD, *n* = 3 independent experiments in (**A**–**E**) and 2 independent experiments in (**F**–**I**). Two-way ANOVA followed by Tukey’s multiple comparisons test. #, comparison between F and R in a condition; £, comparison between F and C in a condition; and *, comparison of C in a treated condition with the C of control condition.

**Figure 10 biomolecules-13-00319-f010:**
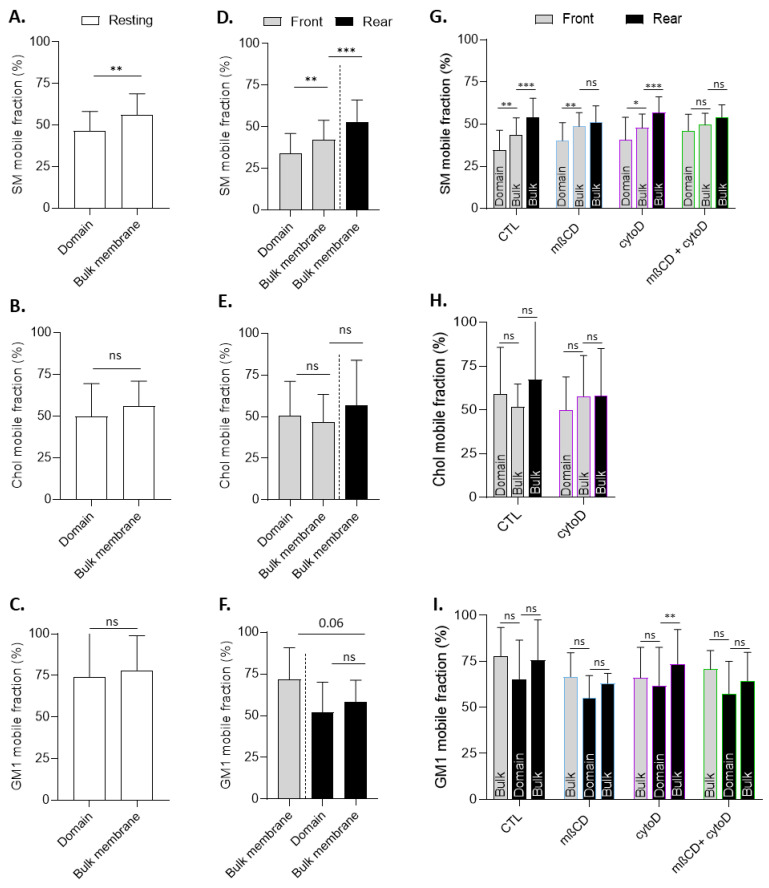
The lateral diffusion of sphingomyelin associated with domains is specifically restricted at the migration front in a cholesterol- and cytoskeleton-dependent manner. (**A**–**C**) Resting cells were single-labeled for SM (BODIPY-SM; **A**), chol (theta; **B**), or GM1 (CTxB; **C**), then lateral diffusion was evaluated by FRAP on 2 regions of interest: lipid domain or bulk membrane. (**D**–**F**) Cells that migrated for 5 h in Ibidi chambers were single-labeled for SM (**D**), chol (**E**), or GM1 (**F**) as in (**A**–**C**), then lateral diffusion was evaluated by FRAP on 3 regions of interest, bulk membrane at the front, bulk membrane at the rear, and lipid domain either at the front (SM, chol) or at the rear (GM1). (**G**–**I**) Cells were left untreated or pretreated with 5 mM mβCD, 0.5 µM cytoD or a combination of both treatments (except for panel H since theta labeling was strongly reduced upon mβCD), then migrated for 5 h in Ibidi chambers. Cells were next single-labeled for SM (**G**), chol (**H**), or GM1 (**I**) and analyzed for lipid lateral diffusion as in (**D**–**F**). The data are expressed as the % of mobile fraction (means ± SD, *n* = 21–33 cells for each condition from 5 independent experiments). Paired *t*-test, one-way ANOVA, and two-way ANOVA, followed by Tukey’s multiple comparisons test. The statistics between different groups are indicated with bars on top of the graphs.

**Figure 11 biomolecules-13-00319-f011:**
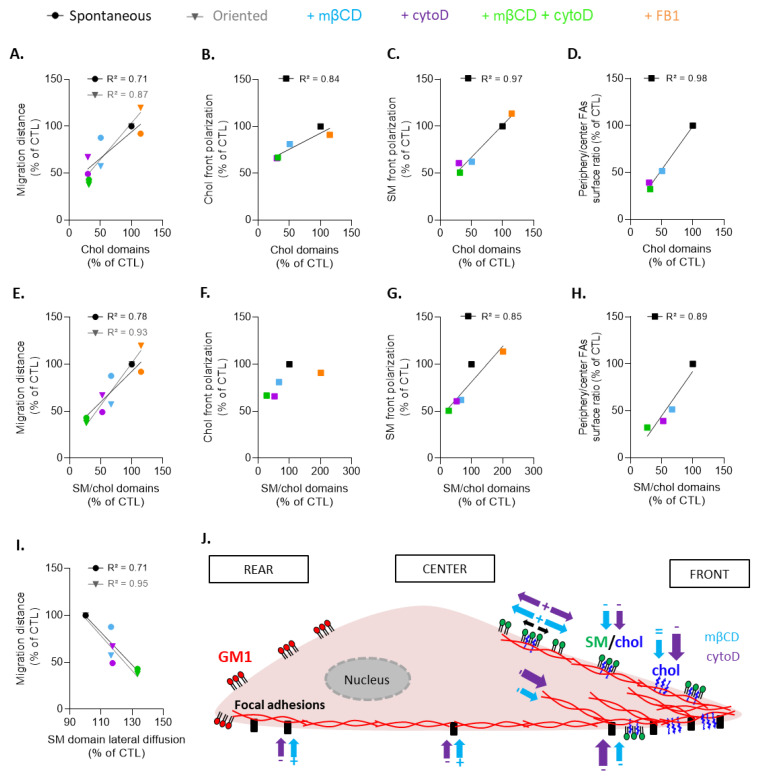
Relations between myoblast migration, polarization of cholesterol and sphingomyelin at the front or focal adhesions peripheral distribution on one hand, and chol- and SM/chol-enriched domain proportion or SM lateral diffusion on the other hand, and hypothetical model. (**A**–**H**) Positive linear correlations of chol-enriched domains (**A**–**D**) or SM/chol-enriched domains (**E**–**H**) with myoblast spontaneous (black circles) and oriented (gray triangles) migration (**A**,**E**), chol and SM polarization at the migration front (**B**,**C**,**F**,**G**) and the ratio of periphery vs. center FAs surface occupancy (**D**,**H**). (**I**) Negative linear correlation of SM-enriched domains lateral diffusion with myoblast spontaneous and oriented migration. Regression and R^2^ are indicated on the graph only if >0.5. (**J**) Hypothetical model. While GM1-enriched domains polarized at the rear (red), chol (dark blue)- and SM/chol (green/dark blue)-enriched domains polarized at the front in proximity with the actin cytoskeleton (red) and SM-enriched domains presented a restricted lateral mobility (black arrows). Chol depletion by mβCD impaired SM polarization as well as F-actin and FA (black rectangles) polarization in favor of a distribution at the center and rear of the cell (blue arrows). Actin cytoskeleton impairment by cytoD abrogated the effects of mβCD on SM polarization and FA distribution (purple arrows). In contrast, the combination of mβCD and cytoD increased the lateral mobility of SM associated with domains (purple and blue arrows). Size of arrow reflects the extent of the effect: (+), increase; (−), decrease; (=), no effect.

## Data Availability

Not applicable.

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
