# Peer review of "Cholesterol and Sphingomyelin Polarize at the Leading Edge of Migrating Myoblasts and Involve Their Clustering in Submicrometric Domains"

_biomolecules, 2023, doi:10.3390/biom13020319_

Round 1

Reviewer 1 Report (Previous Reviewer 2)

The authors have made the requested improvements; the manuscript is more complete and flowing.

Author Response

Reviewer 2 Report (New Reviewer)

Manuscript biomolecules-2176342

Cholesterol and sphingomyelin polarize at the leading edge of migrating myoblasts and involve their clustering in submicrometric domains by Vanderroost et al.

Authors studied whether SM, chol and GM1 can cluster into lipid domains at the myoblast cell surface. They demonstrated the clustering in chol- and SM/chol-enriched microdomains providie a mechanism for the spatial and functional control of myoblast migration.

The work is well-conducted, methodology is adequate, and the presentation of results is outstanding. All together I can say it is a nice piece of research. 

Author Response

Reviewer 3 Report (New Reviewer)

This is an interesting manuscript examining in detail the cellular distribution of cholesterol (Chol), sphingomyelin (SM) and GMI in C2C12 myoblasts. Fluorescent lipid analogs and/or toxin fragments were used to analyze resting as well as migrating cells. The utilization of several imaging techniques in combination with a variety of experimental tools, such as cholesterol depleting agents, cytochalasin D and sphingoIipid synthesis inhibitor FB1 provides a detailed characterization of domains that contain different amounts and combinations of SM, Chol or GMI in resting cells vs the front and rear of myoblasts during migration, together with their association with the actin cytoskeleton. The work supports previous concepts developed by others and extends the overall understanding of the differential roles and distribution of Chol, SM and GMI in membrane microdomains during cell migration, using myoblasts as a model. There are a few points that should be addressed:

1.     The Discussion is extremely long and is currently more than 5 pages in length. While this demonstrates an attention to detail when discussing the results, this becomes exhausting for the reader. The authors should focus on the most relevant points and drastically condense the content in this section.

2.     In several figures it is stated that some data sets were derived from a single (1) independent experiment, which probably is reflected by the lack of SD for some values in Fig 3C, 6G, 7G-H, Suppl Fig 2B-C, Suppl. Fig. S4. Does this mean data from only a single cell is shown? The number of cells analyzed should be provided and if more than one cell was analyzed, a mean+/-SD can be calculated, even if this was derived from a single experiment.

Minor

1.     Please provide a reference (e.g. review article) for SM, Chol and GMI being mainly and/or exclusively located at the outer PM leaflet

2.     Figure 2: How were the lines for the fluorescent intensity profiles selected, e.g. points 1-6 in the enlarged inlet?

Author Response

Reviewer 4 Report (New Reviewer)

The authors investigated the contribution of plasma membrane cholesterol and sphingolipids to myoblast migration during myogenesis and muscle tissue homeostasis. It is a well-designed work.

1. The discussion section is too long. Authors should shorten this section by at least 20% and avoid repetition.

Author Response

This manuscript is a resubmission of an earlier submission. The following is a list of the peer review reports and author responses from that submission.

Round 1

Reviewer 1 Report

This manuscript focuses on the involvement of lipid organization of the plasma membrane in cell polarization and migration. In particular, the authors are interested in the participation of sphingolipid and cholesterol enriched microdomains, also called lipid rafts. This subject has been extensively investigated in the early 2000s in different cell models.

Previous publications of the team reflect a clear expertise on the characterization of lipid microdomains by confocal microscopy (airyscan) and AFM approaches. This expertise is applied in this study in a new cell model that are C2C12 murine myoblasts.

The first part shows in a convincing and original way in resting myoblasts the likely existence of different classes of microdomains variously enriched in sphingomyelin, GM1, Cholesterol (Fig2C). Under migration conditions, the proportion of each class of these microdomains seems to be modified. Interestingly, the distribution of the different classes of microdomains does not seem to be identical between the front and back poles of the polarized cells (Fig4B). The author claim that those enriched in cholesterol are more located at the front of the cell while those enriched in GM1 are located at the back. It would be interesting to determine if the total number of microdomains relative to the cell surface changes between resting and migrating conditions. We only have access to data indicating the proportions of the different categories of microdomains.

It is also unclear whether migrating cells were also placed at 4°C before labelling. If so, it might impact significantly on their polarized morphology depending on how long they’ve been placed at this temperature. I recommend the author compare and show the morphology of the migrating cells fixed at 37°C or fixed after placing them for several minutes at 4°C to verify whether the morphology is not affected after a period at cold temperature.

From the methods described, it appears that the labeling of lipids was performed on cells previously placed at 4°C. Considering the influence that low temperatures can have on the formation of microdomains, one can question whether all the microdomains observed exist also under conditions of physiological temperature of 37°C. It would be interesting to perform additional experiment to verify whether identical microdomains exist at 37°C. On the other hand, the cholesterol labeling is only done with theta toxin. It would be interesting to use other markers like filipin or fluorescent cholesterol (TOP Fluor cholesterol). In addition, the authors use alternatively for the labelling of sphyngomyelins the mcherry -Lysenin toxin or the BODIPY-SM, can they show more clearly in the C2C12 model that the labelling is identical with the two probes and give the same results. Somehow in the figure legends it looks like the author combined in their quantification the results obtained with both type of SM staining. The use of these different probes (except Filipin) at 37°C would give essential indications on the dynamic of these microdomains, and moreover on the biological relevance of their existence.

If this first part seems interesting and brings something new about the possible existence of different categories of microdomains, the rest of the study leaves me perplexed.

As expected, the objectives of the study were then to characterize:

1) on what does the formation of these distinct microdomains depend?

2) the functional importance of these microdomains on cell polarization and migration as well as on the organization of the actin cytoskeleton and focal adhesions.

To answer these two questions, it is clear that a wide variety of experiments and many measurements have been performed. However, the robustness of some of the data is highly questionable. While the number of times the experiments were performed is indicated (sometimes only 1 or 2) in all legends, the number of cells that were analyzed for each condition in each repeated experiment is not clear. In many cases, the number of cells analyzed seems small enough (Example Fig S1A and B, Fig5 - 9) to be sure that the measured differences are significant. In addition, it is mentioned in all figures that statistical tests were performed. However, in many panels it is unclear whether the results from test performed indicate a significant difference (examples: Fig 8Fand G, Fig 9C).

In some figures (Fig 10, Fig S4) a lot of parameters were extracted from the experiments. It is difficult from the manuscript to understand clearly the conclusion they withdraw.

The paragraph 3.7 must be remodel or be supported by more robust data. Indeed, the author claim that Cholesterol depletion decreases F-actin but increases focal adhesion size and cell occupancy specifically at the center of resting myoblasts”. But as they mentioned themselves, measurements just indicate a tendency without statistical significance.

For both questions, the strategy used by the authors are mainly based on the use of MbCD, well known to be powerful to deplete cholesterol (and other lipids as described in the literature) as illustrated in Figure S2 where 5 mM of MbCD (for 30 min) deplete more than 50 % of membrane and total cholesterol. It has to be noticed that in some experiments, the author had to use 10 mM of MBCD to see an effect (probably significant) on what they measure (example F-actin distribution or FA size and distribution (Fig 8 and 9).

Of course, under these massive depletion of cholesterol (event at 5 mM), cholesterol containing microdomains are affected (Fig 3). We do know what proportion of cholesterol (relatively to the total cellular cholesterol) is present in these plasma membrane microdomains. However, we can assume that it is far below 50 %. In these conditions, after MbCD treatment (at the high dose of MBCD used and longtime treatment), all the inhibitory effects observed on migration and polarization and the changes observed on Focal adhesions and actin cytoskeleton could not be attributed only to the effect of cholesterol containing microdomains disruption. Cholesterol also influence the behavior of many intracellular vesicles that are involved in regulating integrin dependent cell adhesion actin dynamic.

In addition, massive cholesterol depletion could induce a stress leading to cell retraction leading to FA disassembly at the periphery. Thus, the observed effect may not be related only to the decrease in the proportion of cholesterol containing microdomains.

In this context, a lot of conclusions made by the authors regarding the role of the subpopulation of cholesterol microdomains (SM/Chol, GM1/Chol, SM/Chol/ GM1) at the front edge, on cell polarity establishment and migration appear abusive. They are mainly based on correlated observations showing that cholesterol depletion modifies these microdomains and also has an impact on cell polarity, migration, AF and F-actin properties.

To better answer these questions, the authors should use better tools or strategies to specifically disrupt only the "likely to exist" microdomain subtypes they identify.

In the discussion of the manuscript, it is difficult to understand how at the molecular level, the authors imagine the microdomains they identify would play a role. The authors must improve this point  by placing their results in the context of the literature describing the impact of cholesterol on cell adhesion (i.e. Ramprasad et al. https://doi.org/10.1002/cm.20176 ; Takahashi et al., DOI: 10.15252/embj.2020106871 , …)

In conclusion, I think that the first part of the present study can provide interesting new information in the field of membrane microdomains if additional experiments are performed (as suggested). But the second part of the study is based on data that are not robust enough and experimental approaches that lack too much specificity. I suggest further exploration of the molecular mechanisms driven by the described cholesterol microdomains involved in the dynamic organization of F-actin and integrin-dependent structures.

Reviewer 2 Report

In the manuscript entitled: “Cholesterol polarization at the leading edge is required for myoblast migration and involves its clustering in submicrometric domains”, the aims of the study by Vanderroost et al., were to determine whether lipid domains could be found on the surface of the myoblast and their implication in the spontaneous and oriented migration of myoblasts. For this purpose, the authors use the already well-characterized line of murine myoblasts C2C12.

The structure of the proposed study is very clear; the methods are accurate and precise; all the tables and figures are instrumental for the content of the manuscript.

Although the experimental work is linear and complete, few notes need to be addressed.

Minor Revision

1.     In my opinion is not necessary to use **** for p< 0.0001. I recommend using the following legend: *p< 0.05; **p< 0.01; ***p<0.0001.

2.     In order to have a more immediate understanding of the experiment, it would be necessary to modify the A, B, C panels in figure 3: the concentrations reported on the bars are confounding;

3.     In figure 7, you need to review the statistic in the graphs: the significance on the bars is difficult to understand;

4.     The statistical part is well written in the “Statistical analysis” paragraph, but it is too concise in the legends of figures where a more accurate description is necessary.

Mainly, this manuscript is appropriate for the publication in Biomolecules after providing proper replies to the minor comments suggested.